# Cognitive–Linguistic Profiles of German Adults with Dyslexia

**DOI:** 10.3390/bs15040522

**Published:** 2025-04-13

**Authors:** Linda Eckert, Gesa Hartwigsen, Sabrina Turker

**Affiliations:** 1Research Group Cognition and Plasticity, Max Planck Institute for Human Cognitive and Brain Sciences, 04103 Leipzig, Germany; eckert@cbs.mpg.de (L.E.); hartwigsen@cbs.mpg.de (G.H.); 2Wilhelm Wundt Institute for Psychology, Leipzig University, 04109 Leipzig, Germany; 3Brain and Language Lab, Department of Behavioral and Cognitive Biology, University of Vienna, 1030 Vienna, Austria

**Keywords:** dyslexia, orthography, reading, German, adults, language, phonological awareness

## Abstract

Past research has extensively explored reading in English-speaking children with dyslexia who acquire a highly irregular and opaque orthography. Far less is known about the manifestation of dyslexia in shallow, highly consistent orthographies like German, especially in adults. To shed further light on the heterogenous manifestation of dyslexia in German-speaking adults, we assessed reading and reading-related abilities, spelling, cognitive abilities, and language learning experience in 33 healthy German-speaking adults (17 females) and 33 adults with dyslexia (20 females). The four main aims were to (1) elucidate the intricate relationship between cognitive and literacy abilities, (2) investigate persisting weaknesses, (3) determine the strongest predictors of dyslexia, and (4) investigate deficit profiles. Group comparisons revealed persistent deficits in almost all measures of reading and spelling, slight deficits in verbal working memory, but no visuospatial impairments in adults with dyslexia. Moreover, adults with dyslexia had considerably lower English skills and lower educational attainment. Overall, we found fewer and weaker links between literacy and cognitive measures in adults with dyslexia, indicating a dissociation between these skills. Spelling, word reading, and phonological awareness were the best predictors of dyslexia, but the most widespread deficit was rapid automatized naming. Our findings suggest a heterogeneous manifestation of dyslexia in German-speaking adults, with even low-level deficits persisting into adulthood despite the shallow nature of the German orthographic system.

## 1. Introduction

Developmental dyslexia is a learning disorder that manifests in the form of severe deficits in reading and writing ([59]). According to recent prevalence studies, it affects at least two to three children in an average classroom ([28]; [60]) and deficits in reading and writing often co-occur with related cognitive deficits ([16]; [23]). These comprise weaknesses in phonological awareness (PA), lexical access, executive functioning (e.g., working memory/WM), attention, and auditory and visual perception (e.g., [7]; [25]; [55]). Deficits frequently persist into adulthood and may result in lower levels of mental well-being, suggesting that while individuals with dyslexia often meet external markers of success, they may suffer higher personal and psychological strain when doing so ([58]).

Most research in the field of dyslexia has been performed with speakers of English and the focus has been mainly on children. As such, conclusions on the nature of dyslexia are largely based on its manifestation in English and in children during literacy acquisition. Although cognitive and pre-literacy skills are largely similar across languages ([33]), variations in the writing system are known to affect literacy acquisition. Overall, alphabetic orthographies can be categorized on a continuum from shallow to deep consistency based on their syllabic complexity and the nature of orthographic (i.e., sound to letter) mappings ([49]). Whereas German and English are both alphabetic languages with a similar orthographic system and phonology, they have distinct orthographic consistencies ([61]). English, with its irregular and inconsistent grapheme–phoneme correspondences, is commonly known as having a deep or opaque orthography ([49]). For instance, the pronunciation of the graphemes ‘ou’ follows no specific rule: [aʊ] in ‘loud’, [oʊ] in ‘soul’, [uː] in ‘soup’, [ʌ] in ‘touch’, [ʊ] in ‘would’, [ɔː] in ‘four’, [ɜː] in ‘journal’, [ɒ] in ‘rough’, and [ə] in adjective endings ‘-ous’ (‘glamorous’, ‘famous’). German, in contrast, has a rather shallow orthography with highly consistent letter–phoneme combinations and few exceptions ([21]). As such, most letters are assigned to single sounds and exceptions usually follow simple rules. A good example of such an exception is the reduction of the letter ‘e’ in a non-accentuated syllable at the end of a word: [e] always becomes [ə] (so-called schwa), e.g., ‘viele’ [ˈfiːlə], ‘andere’ [ˈandəʁə]. Consequently, English and German differ in the extent to which the spelling of a word can be used to predict its pronunciation, which is crucial for learning to read fluently.

In the past few years, more and more studies have started to address the issue of orthographic depth and potential deficits in literacy and cognitive abilities in shallow languages. Single studies include [22] ([22]) who observed a high prevalence of decoding problems in Croatian-speaking children (three out of five profiles), despite Croatian being a highly shallow language with few irregularities. Recently, a meta-analysis ([45]) and a meta-analytic study ([6]) explored the effects of orthographic depth on the reading and spelling skills of individuals with dyslexia. [45] ([45]) explored reading skills in adults with dyslexia across alphabetic languages, focusing on (1) reading and writing, (2) reading- and writing-related skills [e.g., PA, rapid automatized naming (RAN)], and (3) cognition. They found that deficits in reading accuracy, word and nonword reading, spelling, and PA were less pronounced in more shallow languages, suggesting a crucial moderating role of orthographic depth on dyslexia profiles and outcomes in adulthood (see also [51]). [6] ([6]) reviewed 113 studies and added age and orthographies as moderators to analyses, uncovering an age-by-orthography interaction on word reading accuracy but no effect of orthographic depth on fluency parameters. While these two works have some shortcomings, e.g., the coding of orthographic depth, subsuming children and adults, and the inclusion of non-alphabetic languages (e.g., Hebrew), they provide strong evidence for the orthographic depth of a language playing a significant role on the manifestation of dyslexia across the lifespan.

Regarding cognitive and reading profiles in adults with dyslexia, research suggests that primarily deficits in reading speed and writing persist into adulthood ([15]; [37]). Highly educated adults with dyslexia have been shown to develop successful coping strategies (e.g., [38]), suggesting that educational attainment may influence the occurrence or severity of reading- and writing-related weaknesses in adults with dyslexia. Although most evidence comes again from studies with English speakers, a few studies explored cognitive-linguistic profiles of adults with dyslexia in other languages. Reid and colleagues uncovered that most Polish individuals with dyslexia exhibited a phonological deficit, and a small subset had a cerebellar or visual magnocellular deficit ([44]). Nergard-Nilssen and Hulme found that spelling problems were the most persistent deficit in Norwegian adults with dyslexia, with weaknesses in phoneme awareness, RAN, and WM showing strong links to literacy problems ([36]). A similar pattern emerged for Dutch adults with dyslexia, where most participants shared a phonological core deficit, but large variability was uncovered in orthographic coding (i.e., accessing a printed word from memory) ([1]). Interestingly, although German is usually named as an example of a shallow orthographic system, no behavioral studies have investigated cognitive–linguistic profiles in German-speaking adults with dyslexia to our knowledge. Various studies with German-speaking children with dyslexia ([26], [25]; [63]) suggest that in comparison to English-speaking children, German-speaking children with dyslexia are less accurate and slower during pseudoword/nonword and low-frequency word reading, with PA and RAN being the best predictors of phonological deficits. However, this link has not been confirmed in adults.

To summarize, the cognitive–linguistic profiles of adults with dyslexia have been neglected in dyslexia research to date and most research was performed with English-speaking populations. Owing to this gap, it remains unclear whether and to what extent dyslexia affects reading and writing and associated cognitive abilities to varying degrees in adulthood in German speakers. The four main aims of the present work were thus as follows: (1) elucidate the intricate link between cognitive and literacy abilities, (2) explore persisting weaknesses, (3) determine the strongest predictors of dyslexia status, and (4) investigate deficit profiles in RAN, PA, and working memory (WM) in adults with dyslexia. Based on previous works comparing deep and shallow languages, we hypothesized that German-speaking adults with dyslexia have marginal problems with phonological decoding and awareness due to the consistent and regular sound-letter mappings in German. In comparison to typical readers, they should thus show little to no weaknesses in low-level reading skills (e.g., access to letters, digits, or PA) but rather in high-level reading and writing, including word and text reading speed. In addition to investigating these hypotheses, we also looked at deficit profiles in an exploratory fashion to uncover a potential persistent phonological deficit and the occurrence of cognitive weaknesses, such as verbal and visuospatial WM deficits in German-speaking adults with dyslexia. Additionally, we aimed to uncover the best predictors of dyslexia status, which we hypothesized to differ substantially from children, who display primarily deficits in PA and RAN.

## 2. Materials and Methods

### 2.1. Participants

We tested a group of 33 young healthy German-speaking adults (control group/CG, 16 males/17 females; age range: 21–39 y, Mage = 28.4 ± 5.0 y) and 33 young German-speaking adults with dyslexia (13 males/20 females; age range: 18–39 y, Mage = 25.6 ± 5.8 y). All participants with dyslexia were either officially diagnosed with dyslexia as a child or had exhibited literacy problems over the years with a present performance at least one standard deviation below the mean of at least 50% of administered reading and spelling tests (please note that this only affected two included subjects). No subjects had any known comorbidities with other learning or developmental disorders. All participants had nonverbal intelligence scores within the normal range or above (nonverbal IQ: ≥91; CFT 20-R; Weiß, 2019). All testing took place at the Max Planck Institute for Human Cognitive and Brain Science in Leipzig. Participants received a monetary reward for their participation in the study. Prior to participation, written informed consent was obtained from each participant as approved by the local ethics committee of the University of Leipzig under the guidelines of the Declaration of Helsinki. Please note that the present sample is an extended sample from a previously published neurostimulation study with typical adult readers and adults with dyslexia ([53]). Therefore, the sample size was initially determined based on comparable previous neurostimulation studies, and no initial power analysis was performed. However, to confirm that the chosen sample size was sensitive enough to detect the expected effect sizes, we computed a post hoc sensitivity calculation with G*Power ([9]). Assuming α = 0.05, we had 80% power to detect effect sizes larger than 0.717 (Effect size *d*) or 0.338 (Effect size *r*) for non-parametric two-tailed *t*-tests with the respective group sizes (*N* = 33 per group).

### 2.2. Assessment of Reading and Reading-Related Abilities

In terms of reading, we assessed word reading, pseudoword/nonword reading, and text reading. For word and pseudoword/nonword reading, the one-minute Salzburger Lese- und Rechtschreibtest II (SLRT-II; [32]), testing reading speed and accuracy of both words and nonwords, was used. Next, the ability to read text was tested using the Lesegeschwindigkeits- und Verständnistest für die Klassen 5–12+ (LGVT 5–12+; [47]). In this test, participants have six minutes to read a text with missing blanks which have to be filled by selecting from three options based on context. Standardized scores were calculated for reading speed (number of words read in six minutes), reading accuracy (ratio of filled gaps and correct items), and reading comprehension scores (number of correctly inserted words).

For an assessment of spelling, participants completed the Rechtschreibungstest (RT; [20]). This test is a dictation test and requires filling in the missing words in a German written text that was read aloud to them. The number of incorrectly written words was converted into percentages, reflecting participants’ ability to spell.

PA was assessed using a spoonerism task (German adaption of [39]). Here, study participants were asked to exchange the first sounds of the first name and the surname of 12 well-known German personalities and characters. The inner, silent processing time and the time needed to say the correct words out loud were determined and labeled as reaction time (RT) and production time (PT). Moreover, RAN was assessed by the Rapid Automatized Naming Test in TEPHOBE ([31]). Participants had to correctly name sets of colors, digits, and letters as quickly as possible, and the raw score was computed by calculating the number of items named per second. All assessments for both reading and reading-related abilities as well as cognitive and language-related assessment are summarized in Table 1 (for the concrete references, see Section 2.2 and Section 2.3).

### 2.3. Cognitive and Language-Related Assessment

The short form of the Culture Fair Test Scale 2 Revision (CFT 20-R; [56]) was used to assess nonverbal intelligence (henceforth IQ). This standardized test contains four subtests (continuation of series, classification, matrices, and topological conclusions) that have to be completed with time limits (3 vs. 4 min). During the short training phase, subjects received instructions on the specific task of the subtest and had the chance to ask questions. Raw scores were transformed into age standards.

Subjects performed three verbal, phonological WM tests, namely digit span forward, digit span backward, and nonword span. The stimuli for both digit span tests were taken from the Wechsler Adult Intelligence Scale (WAIS-IV; [40]). Participants had two trials to accurately repeat the digits before another element was introduced. They received one point for each correct repetition of strings, with a maximum of 14 points attainable in each task. For nonword span, the Mottier Test was used ([35]), where blocks of nonword syllables had to be repeated. The test was terminated when subjects could repeat <50% of the nonword syllables. All three measures formed the composite variable verbal WM (VWM) for the discriminant analysis due to Cronbach’s alpha confirming high reliability among them (α = 0.84).

To assess visuospatial WM, participants completed a computer-based version of the Corsi Block-Tapping Test ([8]) for both correct order (forward condition) and reverse order (backward condition). Subjects always had two attempts to click the specific number of blocks in either forward or backward order.

Regarding language skills and education, subjects were required to fill out a registration form for the participant database of the Max Planck Institute for Human Cognitive and Brain Sciences. In addition to providing information on their highest education, they had to indicate their self-rated foreign language proficiency in English (their second language) on a five-point Likert scale, ranging from (1) very good in spoken and written language to (5) beginner’s knowledge. All subjects had learned English as their second language from age 8 on at the respective schools.

### 2.4. Statistical Analyses

All statistical calculations were performed using JASP 0.17.3 ([52]) and SPSS 27 ([18]). First, Bayesian correlations were computed in JASP 0.17.3 for each group separately to explore links between reading and cognitive abilities in the two groups (typical readers vs. adults with dyslexia). We only report robust findings with a Bayesian factor (BF_10_) > 10, which signifies robust correlations ([27]). Following the correlational analyses, group differences in behavioral performance between typical readers and adults with dyslexia were assessed using the non-parametric Mann–Whitney U test as performance in the dyslexia group was largely not normally distributed as assessed through the Shapiro–Wilk test in JASP 0.17.3. Please note that JASP provides the Mann–Whitney U-statistic as *W* since it is an adaptation of Wilcoxon’s signed-rank test ([13]). As a measure of effect sizes, the rank biserial correlation coefficient *r* is provided. Similarly to Person’s *r*, effect sizes of *r* > 0.3 are considered as large effects ([12]; [13]). We adjusted *p*-values for multiple comparisons using Holm correction with the p-adjust function in RStudio version 4.3.1 ([41]).

To explore the factors that could best distinguish typical readers from adults with dyslexia, we carried out a discriminant analysis including all administered reading, spelling, and cognitive tests (word reading, nonword reading, spelling, digit span forward, digit span backward, nonword span, text reading comprehension, text reading speed, Corsi forward, Corsi backward, RAN digits, RAN letters, RAN colors) in SPSS 27. Combined structural matrices were computed and correlations with the respective discriminant functions below *r* = 0.3 were not taken into consideration.

Finally, to further investigate corresponding cognitive-linguistic profiles, different types of deficits across the dyslexia group were analyzed. For this, findings from the correlational analyses, as well as Cronbach’s alpha, were used to calculate composite scores for verbal WM, visual WM, and RAN. Subjects were classified as impaired in the respective area if a cut-off criterion of −1.5 SDs below the mean of the typical readers was achieved. The significance level was set at *p* < 0.05.

## 3. Results

The present study aimed to elucidate the link between cognitive and literacy abilities, explore persisting weaknesses, determine the strongest predictors of dyslexia status, and investigate deficit profiles in RAN, PA, and WM in German-speaking adults with dyslexia. To do so, we first investigated the link between reading(-related) and cognitive measures in typical readers and adults with dyslexia. In the next step, we compared the two groups to explore which specific abilities would show significant differences in our adult sample. Consequently, we elucidated which ability was the best predictor of dyslexia status and explored deficit profiles based on earlier accounts reporting deficits in RAN, PA, and cognitive skills (verbal and visuospatial WM).

### 3.1. Correlational Analyses

Overall, typical readers and adults with dyslexia showed very similar patterns of correlations, especially between literacy measures. However, these were always weaker in the dyslexia group, which showed overall less consistency in ability profiles in terms of literacy and cognitive abilities. As displayed in Figure 1a, typical readers showed strong links between different literacy measures, especially between word reading, nonword reading, and RAN. Word reading was not only linked to nonword reading (τ = 0.528, BF10 = 1312.8) but also to RAN digits (τ = 0.415, BF_10_ = 58.7) and RAN letters (τ = 0.476, BF_10_ = 336.6). Thus, better lexical access as measured by RAN tasks showed strong links to speeded lexical access. Nonword reading, in contrast, was linked to RAN letters (τ = 0.542, BF10 = 2096.7), reaction times for phoneme substitution (τ = −0.470, BF_10_ = 56.7), and production times for phoneme substitution (τ = −0.458, BF_10_ = 42.8). In other words, better and faster phonological decoding, as measured through speeded nonword reading, was linked to faster phoneme substitution (i.e., higher PA). Last, also naming letters (RAN letters) was positively linked to faster phoneme substitution (τ = −0.386, BF_10_ = 11.2).

In terms of cognitive and language skills in typical readers, verbal WM measures showed robust correlations with one another: digit span forward was linked to digit span backward (τ = 0.553, BF10 = 1580.4) and nonword span (τ = 0.612, BF_10_ = 11,276.3). However, the three verbal WM measures were not linked to any literacy measure. High consistency was found for RAN performance, with RAN colors showing significant correlations with RAN digits (τ = 0.465, BF_10_ = 238.3) and RAN letters (τ = 0.362, BF_10_ = 15.4), as well as RAN digits with RAN letters (τ = 0.562, BF10 = 5782.7). Furthermore, Corsi forward and backward span (τ = 0.442, BF_10_ = 54.2) and phoneme substitution reaction time and production time were linked (τ = 0.426, BF_10_ = 25.6). The number of languages, the level of English, and the educational attainment were not linked to any reading or cognitive measure.

The dyslexia group, in contrast, showed overall much weaker and fewer correlations across literacy measures but strong links between similar assessments and within constructs (see Figure 1b). Correlations were found between speeded word and nonword reading (τ = 0.525, BF10 = 1592.8) and between RAN measures: RAN colors—RAN digits (τ = 0.600, BF_10_ = 24,345), RAN colors—RAN letters (τ = 0.470, BF_10_ = 281.9), RAN digits—RAN letters (τ = 0.602, BF_10_ = 26,198.8). In addition, text reading speed and text reading comprehension (τ = 0.829, BF_10_ = 7.2 × 108), Corsi span forward and backward (τ = 0.437, BF_10_ = 105.5), and reaction and production times for the phoneme substitution task correlated with each other (τ = 0.392, BF_10_ = 23.4). In terms of linguistic experience, skills, and education, higher verbal WM (digit span forward) was linked to learning more foreign languages (τ = 0.398, BF_10_ = 16.9). Considering the link between reading and cognitive skills in dyslexia, word reading was linked to RAN letters (τ = 0.362, BF_10_ = 15.4), and nonword reading was linked to RAN digits (τ = 0.366, BF_10_ = 17) and RAN letters (τ = 0.366, BF_10_ = 17). Phoneme substitution correlated with verbal WM, namely digit span forward (reaction times: τ = −0.367, BF_10_ = 13.4; production times: (τ = −0.374, BF_10_ = 15.5). To conclude, RAN and single-element reading were linked, as well as PA and verbal WM.

### 3.2. Group Comparisons

The direct comparison between groups revealed that typical readers outperformed adults with dyslexia on almost all literacy measures. Significant differences were found for word reading (W = 1013.5, *p* < 0.001), nonword reading (W = 967.5, *p* = < 0.001), RAN digits (W = 797.5, *p* < 0.001), RAN letters (W = 906, *p* < 0.001), phoneme substitution reaction (RT; W = 147.5, *p* = < 0.001), and production times (PT; W = 217, *p* = 0.002). Moreover, significant differences were found for text comprehension (W = 857, *p* = < 0.001), text reading speed (W = 909, *p* = < 0.001), digit span forward (W = 683.5, *p* = 0.006), and backward (W = 779.5, *p* = < 0.001). After correcting for multiple comparisons, the slight differences in age, number of languages, and nonword span did not reach significance. The results of the comparisons are summarized in Table 2 and Figure 2. Note that according to the norms for adults as provided in the SLRT-II, the observed median of word reading in the control group (122) would equal a percentile rank of 63–64, that of the dyslexia group (80), a percentile rank of 6–7. Likewise, the median of word reading in the control group (72) equals a percentile rank of 45–47, and that of the dyslexia group (44), a percentile rank of 8–10.

### 3.3. Best Predictors of Dyslexia

The assessed literacy and cognitive measures allowed for a correct classification of dyslexia status in 98.1% of cases (λ = 0.197, χ = 68.2, df = 16, *p* = 0.001; 98.1% of cases correctly classified; see Table 3). Within the included measures, the most significant scales for group distinction were spelling (r = 0.62), word reading (r = 0.58), and nonword reading (r = 0.53). Out of the cognitive and phonological measures, only RAN letters (r = 0.33) showed a significant, i.e., > 0.3, contribution to the distinction between typical readers and adults with dyslexia.

### 3.4. Cognitive–Linguistic Deficit Profiles in Dyslexia

In the next step, we investigated deficits in respective literacy and cognitive domains in our subjects with dyslexia. In line with previous research, we distinguished between phonological deficits (measured by PA), lexical access deficits (measured by RAN), and WM deficits. On average, most subjects with dyslexia showed severe deficits in RAN (84.9%) and phonological problems as measured by phoneme substitution (54.8%). Comparably few individuals with dyslexia had problems with verbal WM (27.3%) and even fewer in visuo-spatial WM (18.2%). Four subjects in our sample, despite having received a dyslexia diagnosis in childhood and having reported persistent reading and spelling problems into adolescence and adulthood, did not show a severe deficit in any of these reported measures. We would like to emphasize, however, that this does not necessarily mean that those subjects had a typical-like performance. Most likely, their performance did not meet our stringent cut-off criteria of −1.5 SDs below the mean.

When comparing the occurrence of deficits across subjects (Table 4), we found that roughly a third of all adults with dyslexia only had one severe deficit (30.3%). Almost all subjects in this group showed an exclusive RAN deficit. The second third (33.3%) had a double deficit, with problems in RAN and PA co-occurring most frequently. Among those 15.2% of subjects who had three severe deficits, a deficit involving RAN, PA, and verbal WM was the most frequent. Approximately a tenth of the adults with dyslexia showed severe deficits that comprised all domains. Since the deficit groups were too small to run any statistical analyses (e.g., the no deficit group contained only four subjects and the quadruple deficit only three subjects), we could not run further statistical analyses comparing the groups.

## 4. Discussion

The present work aimed to shed further light on the heterogenous manifestation of dyslexia in German-speaking adults. So far, most research has focused on English-speaking populations and children. This is problematic since the orthographic depth of a language influences reading acquisition and the encountered deficits in dyslexia. Additionally, deficits change over time and are compensated for, potentially resulting in very different weaknesses in adults as compared to children and teenagers. The four main aims of the present work were to (1) elucidate the intricate link between cognitive and literacy abilities, (2) explore persisting weaknesses, (3) determine the strongest predictors of dyslexia status, and (4) investigate deficit profiles in RAN, PA, and WM in adults with dyslexia. Please note that it was our assumption that dyslexia might manifest differently in German-speaking adults than in speakers of orthographically deep languages and differently than in German-speaking children. Since we did not directly compare these groups in the present work, we can only link them to what has been suggested by previous research.

Based on German being a shallow language, we hypothesized that (1) reading skills were less reliant on phonological skills (like PA) and not as dependent upon cognitive skills, and (2) German-speaking adults with dyslexia had little to no weaknesses in low-level reading skills (e.g., access to letters, digits or PA) but marked deficits in high-level reading and writing, including word and text reading speed. Simple phoneme–grapheme correspondences in shallow languages facilitate faster learning of letter–sound relationships and decoding skills. This is backed by behavioral research providing evidence for orthographic depth influencing the cognitive underpinnings of typical adult reading ([46]; [62]), the speed of literacy acquisition ([24]; [57]), the prevalence, severity, and symptoms of impaired reading (e.g., dyslexia; [26]), and the impact of cognition on reading ability ([5], [4]). In terms of correlations between literacy measures, we observed few and substantially weaker links between skills in the dyslexia group as compared to a control group. Interestingly, neither cognitive nor linguistic (number of languages, level of English) or educational (educational attainment) factors correlated with reading and spelling abilities. These rather weak links between abilities that should go hand in hand, as confirmed by the group of typical readers, could point towards a very heterogeneous manifestation of dyslexia with large differences in test performance and thus a lack of consistency. Moreover, it confirms that phonology-related skills are not necessarily strongly tied to skills relying on lexical access. Regarding the second hypothesis, we found that German-speaking adults in their twenties and thirties still display severe deficits in various reading and cognitive measures not limited to high-level reading skills. More specifically, we suggest that long-term deficits across all reading measures extend to issues in verbal WM, lower educational attainment, and lower foreign language skills. This supports findings from deep orthographies, such as English, which revealed problems across literary and cognitive skills that persisted long into adulthood ([17]).

When directly comparing the groups, we found that despite normal nonverbal intelligence (and on average even higher IQ scores than the typical readers) and the absence of other learning or developmental disabilities, German-speaking adults with dyslexia performed significantly worse on all assessed literacy measures. This included word and nonword reading, text reading comprehension and speed, spelling, and measures of lexical access (RAN), as well as PA. This is also in line with studies with speakers of deep orthographies suggesting that adults with dyslexia differ from typical readers in terms of reading-related cognitive deficits ([42]; [45]), primarily in phoneme awareness and WM performance ([51]). Interestingly, most of our test instruments stemmed from diagnostic batteries to diagnose children with dyslexia since no adult versions were available. In other words, fifteen to twenty years after the period at which dyslexia is usually diagnosed in Germany (~7 years in first grade), these tests seem to be useful instruments to confirm dyslexia in adulthood. The norms available for the SLRT-II showed that adults with dyslexia scored on average in percentile ranks of 6–7 for word reading and 8–10 for pseudoword reading, undermining the persistent deficits encountered in these two abilities.

As a third aim, we wanted to explore which factors best predicted dyslexia status and were keen to see whether the factors would match the findings of studies with German-speaking school children. As hypothesized, the present study failed to corroborate findings of previous research in German-speaking children with dyslexia ([26], [25]; [63]), where PA and RAN were the best predictors of phonological deficits in dyslexia. According to the results of our discriminant analysis, spelling was by far the strongest predictor of a history of dyslexia, and difficulties did not only persist in the pseudoword and low-frequency word domain but reading speed deficits also affected simple words. While this is not in line with observations in German-speaking children, it corroborates the few existing earlier studies in adults with dyslexia reporting that spelling errors and problems with reading fluency are the most persistent characteristics of dyslexia in adulthood ([3]; [19]; [36]). Previous studies in both deep and shallow writing systems, however, suggested that phoneme awareness, orthographic knowledge, and RAN were the most important predictors of reading and spelling in adulthood ([11]). In German-speaking adults with dyslexia, RAN and phonological problems were still frequently found but they were not sufficient to correctly predict who had dyslexia and who did not. Overall, we confirm the importance of phonological processes in German-speaking adults with dyslexia, countering previous studies claiming that phoneme awareness is not important in shallow languages ([30]). The significantly worse performance on our phoneme substitution task backs evidence of a persistent phonological deficit in dyslexia across orthographies and languages (e.g., [1]; [45]).

Last, we uncovered deficit profiles in dyslexia and provided support for lexical access being the most widespread deficit in German-speaking adults with dyslexia. In the present sample, the most widely represented deficits were deficits in RAN (84.9%) and PA (54.8%). Surprisingly, not even a third of our subjects with dyslexia showed deficits in verbal WM (27.3%) and even fewer had problems with visuospatial WM (18.2%). Several studies and reviews confirmed verbal WM problems in German-speaking children with dyslexia, but it remains debated whether they affect only the phonological loop ([29]; [50]) or the central executive, or both ([10]; [48]). We also provide further evidence that tasks with higher WM demands (e.g., manipulation of input instead of just storing), digit span backward in our study, are more difficult for individuals with dyslexia than simple WM tasks, as previously found in teenagers and young adults with dyslexia ([54]). The fact that very few subjects with dyslexia had visuospatial WM deficits adds to the existing literature, where some studies found differences ([10]), while others did not ([2]). Since most studies exploring visuospatial WM deficits were with German-speaking children, the present study adds to a weak indication of visuospatial WM deficits in dyslexia, with only two out of ten adults with dyslexia displaying severe deficits in this domain. About a tenth of our dyslexia sample showed no severe cognitive deficit (12.1%), which is probably due to the stringent cut-off criterion of 1.5 SDs. As such, this reflects less severe impairments that did not meet this criterion than no literacy and cognitive impairment at all. It could, however, also support earlier findings that educated adults with dyslexia may adopt strategies to cope with the deficits ([14]; [34]; [38]; [43]). All in all, we provide strong evidence for different dyslexia subtypes, with single deficits in RAN and double deficits in RAN and PA occurring most frequently in German-speaking adults with dyslexia. Interestingly, only one subject in the present study had a single deficit in PA. Conversely, a third of our sample had a single deficit in lexical access (RAN). The extent to which this translates to findings in our whole testing battery and whether the groups differ on other cognitive and literacy measures remains to be further explored in larger samples.

Linking our findings together, we suggest that dyslexia does indeed present in different subtypes, which is reflected in the differences in links between literacy abilities and the large variability as observed through our deficit profiles. Although research on deficit profiles has largely ignored spelling abilities, spelling could be a strong feature of dyslexia across orthographies, even in transparent orthographies with simple grapheme–phoneme mappings. In particular, the level of compensation for spelling deficits and their relation to deficits in other key aspects (PA/RAN) in adulthood could be an interesting avenue for future research.

Please note that the present study is not without limitations. First, the samples in the present study are rather small, especially when it comes to specific deficit profiles. This made it impossible for us to directly compare skills and make robust statements about certain deficits found in subsamples only. Moreover, the included participants with dyslexia will have compensated for their reading and spelling deficits in various ways, which we could not directly address in the present study, nor account for in the statistical analyses. Future studies should aim to compute indices of compensation by retrospectively gathering this information and including it in analyses.

## 5. Conclusions

The present study with German-speaking adults with dyslexia provides evidence for a heterogeneous manifestation of dyslexia, even in orthographically shallow languages like German. Overall, we found fewer and much weaker links between reading, reading-related, and cognitive measures in adults with dyslexia. This can be interpreted as a dissociation between these skills. Moreover, spelling, word reading, and phonological awareness were the best predictors of dyslexia, supporting previous research on spelling deficits being the most persistent feature of dyslexia across orthographies. However, looking at established deficit profiles revealed that the most widespread deficits were in rapid lexical access (RAN) and a double deficit in access and phonological decoding (PA). Future research should aim to contrast languages with different orthographic depths in single studies to further unravel manifestations of dyslexia in different languages and across different age groups, e.g., by comparing young, old, and very old (>65 years) individuals with dyslexia. Furthermore, more research is needed with speakers of shallow languages and a meta-analysis or systematic review could summarize the various findings across languages, improving our understanding of how letter-sound mappings influence manifestations of dyslexia across ages and deficit profiles.

## Figures and Tables

**Figure 1 behavsci-15-00522-f001:**
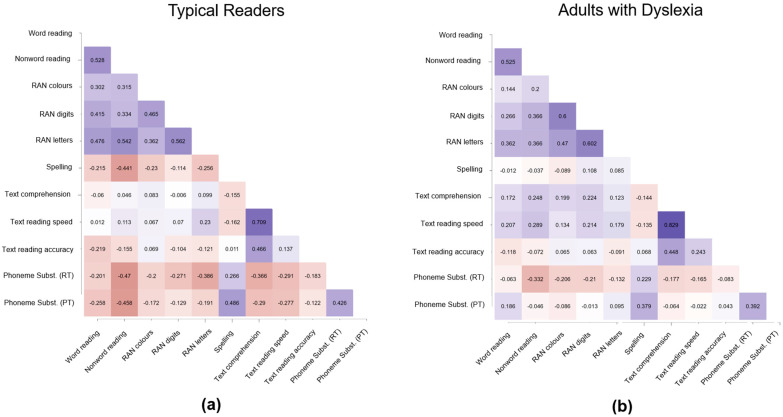
Correlations between literacy measures in (**a**) typical readers and (**b**) adults with dyslexia presented as Kendall’s τ values (blue = positive, red = negative; the darker the blue or red, the stronger/weaker the link).

**Figure 2 behavsci-15-00522-f002:**
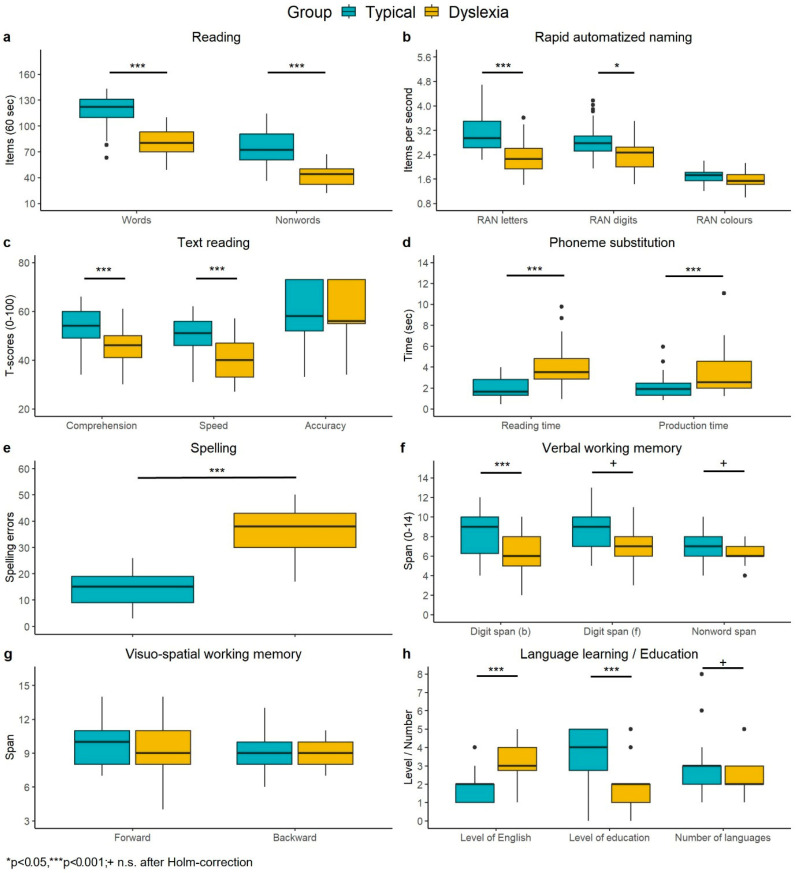
Visualization of group comparisons as presented in Table 2.

**Table 1 behavsci-15-00522-t001:** Summary of reading, cognitive, and language assessments.

Concept/Skill	Specific Task	Assessment Tool
Word reading	Speeded word reading in 60 s	SLRT-II
Nonword reading	Speeded nonword reading in 60 s	SLRT-II
Text reading	Text reading and filling gaps within text (speed, comprehension, accuracy)	LGVT 5–12+
Spelling	Dictation	Rechtschreibungstest
Phonological awareness (PA)	Initial phoneme substitution	Spoonerisms
Rapid Automatized Naming (RAN)	Rapid access to digits, colors, and letters	TEPHOBE
Nonverbal intelligence	Continuation of series, classification, matrices, and topological conclusions	CFT 20-R
Verbal working memory (VWM)	Digit span forward, backward, and nonword span	WAIS-IV, Mottier
Visuospatial working memory (VSWM)	Corsi block span forward and backward	Corsi

**Table 2 behavsci-15-00522-t002:** The direct comparison of behavioral performance between adults with dyslexia (DYS) and typical readers (CG).

Variable	Mdn_cg_ (IQR)	Mdn_dys_ (IQR)	W-Value	*p*	r
Age	29 (7)	24 (8)	722	0.023 +	0.326
IQ	106 (16.5)	112 (18)	408.5	0.236	−0.175
Educational level	4 (2.3)	2 (1)	490	<0.001 ***	0.690
Word reading	122 (21)	80 (23)	1013.5	<0.001 ***	0.861
Nonword reading	72 (30.3)	44 (18)	967.5	<0.001 ***	0.832
Spelling	15 (10)	38 (13)	41	<0.001 ***	−0.925
RAN colors	1.7 (0.3)	1.5 (0.3)	692	0.059	0.271
RAN digits	2.8 (0.9)	2.3 (0.7)	797.5	0.001 *	0.465
RAN letters	3.1 (0.6)	2.4 (0.6)	906	<0.001 *	0.664
Phoneme substitution (RT)	1.6 (1.5)	3.5 (2.0)	147.5	<0.001 *	−0.648
Phoneme substitution (PT)	1.9 (1.2)	2.5 (2.6)	217	0.002 *	−0.481
Text reading comprehension	54 (11)	46 (9)	857	<0.001 ***	0.574
Text reading speed	51 (10)	40 (14)	909	<0.001 ***	0.669
Text reading accuracy	58 (21)	56 (18)	515.5	0.706	−0.053
Digit span (f)	9 (3)	7 (2)	683.5	0.006 +	0.401
Digit span (b)	9 (3.8)	6 (3)	779.5	<0.001 ***	0.575
Nonword span	7 (2)	6 (1)	673.5	0.048 +	0.276
Corsi span (f)	10 (3)	9 (3)	608	0.064	0.271
Corsi span (b)	9 (2)	9 (2)	493	0.840	0.030
Level of English	2 (1)	3 (1.3)	146	<0.001 ***	−0.664
Number of languages	3 (1)	2 (1)	575	0.025 +	0.325

Abbreviations: Medians (Mdn), inter-quartile ranges (IQRs), Wilcoxon rank sum test statistics (W), rank–biserial correlation coefficient (r). + not significant after correction for multiple comparisons (Holm); * <0.05, *** <0.001 (after correction for multiple comparisons).

**Table 3 behavsci-15-00522-t003:** Results of the discriminant analysis showing the contribution of literacy and cognitive variables to correctly classify adults with dyslexia (λ = 0.197, χ = 68.2, df = 16, *p* = 0.001; + at threshold).

Structure Matrix	
Spelling	−0.619
Word reading	0.584
Nonword reading	0.527
RAN letters	0.328
Text reading speed	0.304
Phonological awareness (PA)	−0.297 +
Digit span backward	0.250
Text comprehension	0.243
Digit span forward	0.225
RAN digits	0.222
Phoneme substitution (PT)	−0.176
Nonword span	0.151
Corsi span forward	0.138
RAN colors	0.135
Corsi span backward	0.052
Text reading accuracy	−0.051

**Table 4 behavsci-15-00522-t004:** Frequency of number of deficits for PA, RAN, and WM domains. Subtotal and total frequencies are provided. Additionally, average performances in the respective deficit groups for word reading (WR), nonword reading (NR), text comprehension (RC), text reading speed (RS), and spelling.

Cognitive-Linguistic Deficits	Sub-Total		Total		WR	NR	RC	RS	Spelling
	*N*	%	*N*	%					
No Deficit			4	12.1	92.0	55.0	51.3	46.3	36.5
Single			10	30.3	81.8	44.4	44.3	38.3	36.2
PA	1	3							
RAN	9	27.3							
Double			11	33.3	78.0	41.2	47.2	39.9	32.1
RAN + PA	7	21.2							
RAN + VWM	2	6.1							
RAN + VSWM	2	6.1							
Triple			5	15.2	77.8	39.2	40.8	34.4	37.8
RAN + PA + VWM	4	12.1							
RAN + PA + VSWM	1	3							
Quadruple			3	9.1	69.0	31.7	40.3	39.0	44.7

## Data Availability

All data are openly available at https://osf.io/kh3xj/ (accessed on 20 February 2025).

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
