# Peer review of "Cognitive–Linguistic Profiles of German Adults with Dyslexia"

_behavsci, 2025, doi:10.3390/bs15040522_

Round 1

Reviewer 1 Report

Comments and Suggestions for Authors

Thank you for the opportunity to review this work. The manuscript entitled Cognitive-linguistic profiles of German adults with dyslexia is very clear and nicely written paper that contributes to the dyslexia research community in terms of the language studied, age of the participant groups that were studied and the abundancy of measures included in the analyses.

The paper is clear and concise. I would suggest several minor points that could add to the quality of the paper. Minor suggestions are as follows:

Abstract: Please mention the aims as you mention them elsewhere, i.e. in the Discussion (4 aims).

Intro, pg2 (l. 69). When you mention Reese et al. for the second time, still add the year in brackets (2020)

Intro, pg3 (l. 104-105): Please add the aims exactly as they are added in the Discussion (4 aims) and write the hypotheses as you did elsewhere (discussion).

Materials and methods, pg. 3 (l.142) I presume that one word large is too much. Please delete the final word on the page.

It would be helpful to have a table or a figure with all the variables and tests and tasks used to asses them next to it. Like a scheme a reader could go back to.

Statistical analyses, pg. 5 (l. 197): I just wonder why weren't all analyses performed in one program (like R package)? Is there any particular reason?

Results: this part is very clearly written, and I really like the tables and figures. Maybe it would be helpful to briefly mention the aims in the beginning (pg. 5, l. 226-231) so that your steps clearly correspond to the aims.

Discussion: I am not sure if it would help at this point, but there has been a study on children with dyslexia, that focused on profiling of poor readers in Croatian (also a language with shallow orthography) (Kuvač Kraljević, J., Runje, N., Ružić, V., Škorić, A. M., Lenček, M., & Štefanec, A. (2024). Predictors of reading comprehension and profiling of poor readers in Croatian: educational and clinical perspectives. Frontiers in psychology15, 1297183.)
If you think it could be helpful for the introduction or the discussion.

I am missing the part with the limitation and a firm conclusion to end the study with.

I very much appreciate the opportunity to review this work and I wish you good luck in the next steps.

Reviewer 2 Report

Comments and Suggestions for Authors

The manuscript entitled „Cognitive-linguistic profiles of German adults with dyslexia” investigates reading measures and cognitive skills in German-speaking adults with and without dyslexia. As such, it is one of the few studies investigating reading profiles in adults in a shallow orthography, since most studies’ participants are either children or English-speaking adults. I do have some minor remarks concerning the interpretation of the results and the conclusions of the study. I would ask the authors to address the following points.

General remarks

-As far as I understand, the authors used raw values for the psychometric tests, such as the reading fluency and reading comprehension measures. If so, could the authors also provide percentiles or other standardized values for the tests that do have norm values for adults (e.g. the SLRT-II)? This would be useful to interpret their findings, especially with regard to studies in deep orthographies, such as English (c.f. lines 341-343).

-Could the authors please elaborate how the finding of the most widely represented deficits in RAN is in accordance with the findings that spelling is the best discriminator between adults with and without dyslexia? In general, the four different findings (link between literacy and cognitive profiles, persisting weaknesses in adults with dyslexia, predictors of dyslexia and deficit profiles) should be linked together in the discussion. What do the overall findings of the study mean?

Minor points

-Lines 285-286: Digit span seems to be missing in this list.

-Line 364: Could the authors please clarify how they reach this conclusion?

-There are some typos: line 143: “large” should be deleted; line 327: “deicits” should be “deficits”; line 357: “literary” should be “literacy”; line 364: I suggest deleting “more”
